# An alternative approach for socio–hydrology: case study research

Erik Mostert[1]

[1]Department Water Management, Delft University of Technology, Stevinweg 1, 2628 CN  Delft, the Netherlands

*Correspondence to*: Erik Mostert (e.mostert@tudelft.nl)

**Abstract.** Currently the most popular approach in socio hydrology is to develop coupled human–water models. This article proposes an alternative approach, qualitative case study research, involving a systematic review of 1) the human activities affecting the hydrology in the case, 2) the main human actors, and 3) the main factors influencing the actors and their activities. Moreover, this article presents a case study of the Dommel basin in Belgium and the Netherlands, and compares this with a coupled model of the Kissimmee basin in Florida. In both basins a 'pendulum swing' from water resources

development and control to protection and restoration can be observed. The Dommel case study moreover points to the importance of institutional and financial arrangements, community values and broader social, economic and technical developments. These factors are missing from the Kissimmee model. Generally, case studies can result in a more complete understanding of individual cases than coupled models, and if the cases are selected carefully and compared with previous studies, it is possible to generalise on the basis of them. Case studies also offer more levers for management and facilitate

interdisciplinary cooperation. Coupled models, on the other hand, can be used to generate possible explanations of past developments and quantitative scenarios for future developments. The article concludes that, given the limited attention they currently get and their potential benefits, case studies deserve more attention in socio–hydrology.

## 1 Introduction

Suppose you are interested in how river basins and society interact and evolve together. And suppose you are interested especially in the Great Ouse basin in the east of England. How would you study this basin? One option would be to set up a large interdisciplinary research project to study the topography and geology of the basin and the formation of peat soils since 6000 BC; the human interventions in the Neolithic, Roman and Mediaeval period; the 17th Century drainage works and the resulting peat shrinkage and wastage; the subsequent works to cope with the increasing flooding problems; the changing

governance structure to make these works possible; the changing economy of the basin; and the role of politics and agricultural lobbies (e.g. Godwin, 1978; Richardson et al., 1978; Darby, 1983; Hall and Coles, 1994; Sheail, 2002; Purseglove, 2015; Mostert, 2017a). This would involve extensive literature search and much field work and archival research.

Another option would be to model the co–evolution of water and society. Using insights from previous research on the Great Ouse basin and comparable basins, a coupled human–water model could be developed, which could then be calibrated and validated in order to simulate past and predict future developments. Alternatively, the coupled model could be used as an exploratory toy model to generate possible explanations and develop scenarios for future developments.

The most popular option in socio–hydrology currently is the second one, developing coupled models. In this paper I argue for more attention for the first option, qualitative case study research. I will first briefly review the current socio–hydrological models, focusing on how society is included, and propose a specific type of case study research as an alternative. Next, I will present a case study of the Dommel basin in Belgium and the Netherlands since 1800 and compare this with a coupled model of the Kissimmee basin in Florida (Chen et al., 2016). Both basins are examples of the 'pendulum

swing' (Kandasamy et al., 2014) from water resources development and control to protection and restoration. In the Kissimmee model the pendulum swing is explained in terms of community sensitivity, but I intend to show that important elements are missing from this explanation. In the final section, I will discuss what the best approach for socio–hydrology is. I will discuss the key differences between qualitative case study research and coupled modelling, their comparative advantages and disadvantages, the possibilities to combine them, and whether there are any other approaches that could be

used.

## 2. Socio–hydrological modelling

The term socio–hydrology was coined in 2012 by Sivapalan et al. (2012). It was defined as 'a new science of people and water' that aims at 'understanding the dynamics and co–evolution of coupled human–water system' (Sivapalan et al., 2012, 1271). Socio–hydrology treats society as an endogenous part of the water cycle and studies not only the impact of people on

water, but also of water on people (e.g. Pande and Sivapalan, 2017). This would result in better understanding of long–term developments, better long–term predictions, and better support for water management than approaches that treat society as exogenous, such as scenario–based approaches.

While there are some qualitative socio–hydrological studies (Wescoat, 2013; Gober and Wheater, 2014; Kandasamy et al., 2014; Liu et al., 2014), the dominant approach in socio–hydrology is to develop coupled human–water models. The number

of such models is slowly increasing. The issues modelled include flooding (Di Baldassarre et al., 2013, 2017; Viglione et al., 2014; Grames et al., 2016; Yu et al., 2017; Girons Lopez et al., 2017; Barendrecht et al., 2017); water quality management (Chang et al., 2014); reservoir operation (Garcia, 2016); water supply (Srinivasan, 2015; Ali et al., 2017); groundwater abstraction (Noël and Cai, 2017); land degradation (Elshafei  et al., 2015), subsistence farming (Pande and Savenije, 2016), the pendulum swing from water resources development and control to protection and restoration (Van Emmerik et al., 2014;

Elshafei et al., 2014; Chen et al., 2016; Roobavannan et al., 2017); and the collapse of civilisations (Kuil et al., 2016). Society is included in these different models in different ways. Most commonly, it is modelled as a homogenous actor (e.g. Elshafei et al., 2014; Van Emmerik et al., 2014; Viglione et al., 2014; Grames et al., 2016). In a few cases it is modelled as

one actor consisting of two segments (Chen, et al., 2016; Roobavannan et al., 2017), or as a group of homogenous individuals (e.g. Pande and Savenije, 2016; Noël and Cai, 2017). Management structures and decision–making processes are seldom modelled. The socio–hydrological literature recognises the importance of these issues, but this recognition is not yet reflected in the models made, or only in a rather crude way, for instance in the form a fixed 'cooperativity coefficient'

(Elshafei et al., 2015, 6449). The only exception to date is the model by Yu et al. (2017), which analyses the issue why farmers in Bangladesh are willing to make voluntary contributions to the upkeep of the flood defences when it seems economically rational for them to free–ride on the efforts of others.

Despite their simplifications, most socio–hydrological models can mimic patterns that can be observed in reality. The explanation may be that the models include many variables for which no data are available and consequently have many

degrees of freedom (Troy et al., 2015). It could also be that the variables not included did not vary a lot in the area and period covered. Or it could be a combination of both. Whatever the explanation, the result is that the validity of the models outside of the area and period modelled is unclear and predictions based on them are highly uncertain.

Socio–hydrological systems can never be modelled exhaustively, but it is important to include the most influential variables and processes (cf. Garcia et al., 2016). These depend, first, on the issue of interest, which in turn depends on the disciplinary

background of the researchers and the political and policy context (Lane, 2014). Secondly, they depend on the area and period studied. Societal response to hydrological change may be limited when the costs of action are individual but the benefits collective and when costs have to be made upstream but the benefits are downstream. Societal response will be bigger when there are strong community values (Yu et al., 2017; Mostert, 2017a) and when appropriate institutional arrangements are in place (e.g. Ostrom, 1990; Brondizio et al., 2009). However, such values and arrangements are not

present always and everywhere: they have to be developed and maintained and adapted to changing conditions and needs (e.g. Mostert, 2012).

The socio–hydrological literature contains several recommendations for future research. A first one is public participation (Lane, 2014; Sivapalan and Blöschl, 2015; Srinivasan et al., 2017). Public participation can be a means to obtain data from the public, to educate them, and to promote buy–in of model results and subsequent decisions. In addition, it can be a means

to involve the public in the modelling itself and give them control over what to model exactly and what (policy relevant) assumptions to use.

Another recommendation is to start modelling with clearly defining objectives and to include only the most influential variables and processes given these objectives. This would prevent overly complex models and promote transparency (Garcia et al., 2016). A different recommendation is to move beyond the scale of individual river basins and include more

variables and processes, such as international trade and climate change (Pande and Sivapalan, 2017; Srinivasan et al., 2017). Unless one simplifies in other respects, this results in more complex models.

A general recommendation is to collect more data (e.g. Troy et al., 2015; Blair and Buytaert, 2016). This is essential for calibration and validation and preventing overfitting. Calibration and validation may be less important for exploratory toy models that do not aim to simulate specific systems accurately, but to capture essential processes and feedbacks, generate

possible explanations and explore possible future developments (e.g. Thompson et al., 2013; Di Baldassarre et al., 2015, 2017; Yu et al., 2017). Yet, they should be realistic, and to check this (qualitative) data will be needed.

## 3. Case study research

An alternative approach to developing coupled models is case study research. Case study research can be defined as a qualitative research approach in which a researcher studies one or more systems (cases) within their real-life context through in-depth data collection, using multiple sources of information. The aim is to achieve an in-depth understanding of the system or systems concerned (Yin, 1989; Creswell and Poth, 2017).

In socio-hydrology the relevant systems to study are water and society and their interaction in a specific area, such as a river basin or an aquifer area, and over a long period. The central questions to ask are the following:

1) Which human activities have had a significant impact on the hydrology of the area?

2) Who were the main actors?

3) What were the main factors, hydrological and other, affecting these actors and their activities?

The human activities include all activities that significantly change land use, water use or water flows, such as deforestation, water abstraction, irrigation and drainage, and the construction of reservoirs. The main actors are the main individuals, groups and organizations that:

- use the land and water;

- construct, operate, maintain or finance the infrastructure necessary to use the land and water; or

- regulate land and water use or the infrastructure

Factors that affect these actors and their activities may include the following:

- the values and interests of the main actors;

- the presence of any conflicts between groups, e.g. city versus countryside or upstream versus downstream (e.g. Bavinck et al., 2014);

- the interactions between the different actors (e.g. Pahl-Wostl et al., 2007);

- the presence or absence of a sense community (e.g. Mostert, 2017a);

- the importance of water for the economy in the area (e.g. Roobavannan et al., 2017); and

- the control the actors have over external factors that affect the area's hydrology.

Other factors may be important as well. The trick or 'art' (cf. Savenije, 2009) is to zoom in on those factors that have the most explanatory power in the specific case. Potential explanations as well as alternative explanations should be checked carefully against the collected data. Moreover, as new explanations suggest themselves or new data sources are discovered, the research design may have to be modified ('emergent design': Creswell, 2014, 186).

Depending on the area and period studied, data sources that can be used may include the following:

- archaeological and paleoecological data

- maps
- land use surveys and census data
- the archives of (local) government bodies, water management organisations and other relevant organisations
- laws, byelaws and judicial decisions
- chronicles, topographical descriptions, memoirs and letters
- newspapers
- consultancy reports and other studies
- field surveys
- social surveys
- interviews

The next section will give many examples.

To achieve sufficient detail, it is advisable to focus on one or two important activities, such as the construction of a reservoir. In addition, one could use an existing theory to guide research and help interpretation, such as common pool resources management (e.g. Ostrom, 1990; Araral, 2014) or triple exposure theory (Treuer et al., 2017). The use of theory can be
problematic since most theories focus on specific factors and leave out many others. Consequently, their explanatory power may differ from case to case, and one should always keep an open eye for alternative explanations.

Every case study should finish with some conclusions or 'lessons' that are of more general relevance than the specific case itself, unless the case itself is of special relevance (an 'intrinsic case': Creswell and Poth, 2017). When the conclusions are based on one or a few cases only, they are necessarily tentative, but they may also be based on a comparison with earlier
research. Moreover, they may be tested in subsequent research. If a significant number of case studies have already been published, it is also possible to conduct a meta–analysis, using techniques such as Qualitative Comparison Analysis (Srinivasan et al., 2012).

Cases to study can be selected in different ways (Eisenhardt, 1989; Yin, 1989; Mollinga and Gondhalekar, 2014). A first option is to select a case that is a representative or typical example of the phenomenon of interested, such as the pendulum
swing, in order to explore what are the most important variables and processes. The next section provides an example of this. If there are already quite a few case studies, one can select a very different case to complement the previous research or a very similar case to replicate it, and if there is a theory that seems relevant, one could select a case that is critical for testing that theory (e.g. Aggestam and Sundell, 2016). Practical considerations such as data availability and accessibility should play a role as well. If there is enough time to conduct multiple case studies, one can adopt either a most different or a most similar
design. A most different design involves selecting cases that are as different as possible, to cover as much of the diversity in the population of cases as possible (e.g. Huitema and Meijerink, 2010; Benson et al., 2014). In a most similar design, the cases are as similar as possible except for one variable. This a good approach for studying the effect of an individual variable (e.g. Kochskämper et al., 2016).

Case study research as proposed here is ideally interdisciplinary. Hydrologists should play a large role, but so should historians and social scientists. They are more familiar with many of the data sources that can be used than hydrologists and have specific expertise to contribute. Depending on the area and period studied, other experts should be involved as well, such as archaeologists, soil scientists, ecologists, and river morphologists.

5  **4. The Dommel basin**

To give a better idea of what a socio-hydrological case study may look like and the insights such a study can bring, I will present a specific case: the Dommel River basin since 1800. The Dommel River has its source in Belgium and then flows into the Netherlands until the city of Den Bosch, where it joins the Aa River. Here it changes its name into the Dieze, which after five kilometres discharges into the Meuse River. The Dommel has a basin of circa 1,815 km$^2$, 408 km$^2$ in Belgium and
10  1,407 km$^2$ in the Netherlands (Bongaerts, 1919). The total drop in elevation is 75 metre only, from 77 metre above mean sea level at source, to 25 metre at the Belgian-Dutch border, and 2 metre at Den Bosch. Still, this drop was enough to drive some 30 water mills in the basin.

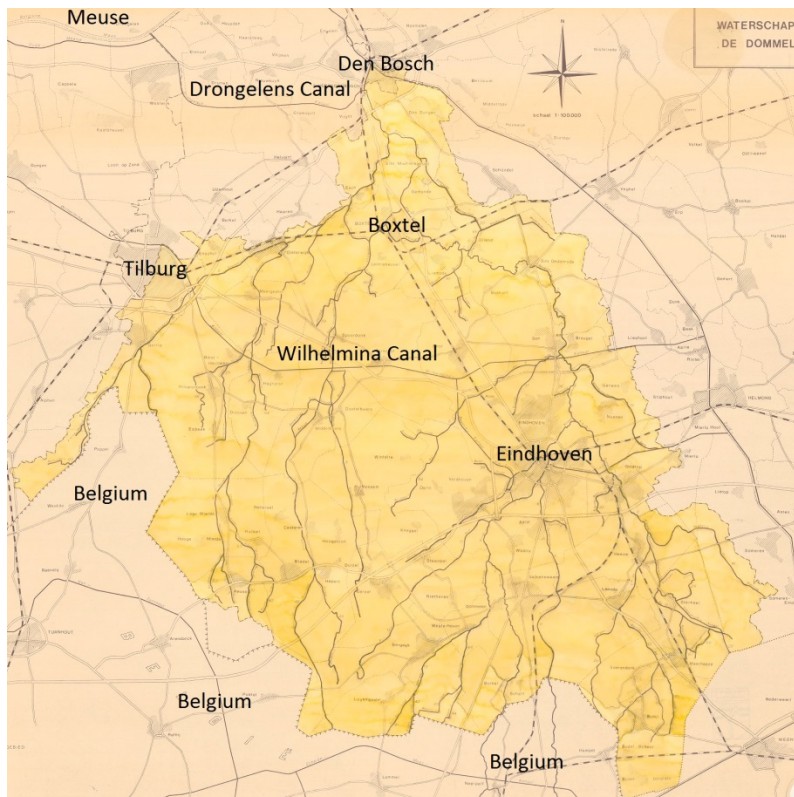

**Figure 1: Map of the Dutch part of the Dommel basin in 1971 (Website Brabant Historisch Informatie Centrum, www.bhic.nl;**
15  **geographical names added)**

The river valleys in the basin used to flood regularly, but until around 1900 only summer floods were seen as problematic as these resulted in a loss of the hay harvest. Winter floods were seen as increasing the fertility of the soil (Bongaerts, 1919; Deckers, 1927; Crijns and Kriellaars, 1987). Dykes could only be found downstream of the last water mill on the river in the town of Boxtel (Fig. 1). Water levels in this part of the river were influenced not only by the upstream discharge, but also by the water level of the Meuse.

From 1875 onwards the Dutch part of the Dommel was regulated to reduce flooding, but from 1990 onwards river restoration projects were executed (Roeffen, 1963; Didderen et al., 2009). This makes the Dommel basin a typical example of the pendulum swing from water resources development and control to protection and restoration.

## 4.1 Modelling the pendulum swing

It would be possible to model the pendulum swing in the Dommel basin in the same way as Chen et al. (2016) have done for the Kissimmee basin in Florida. In their model 'community sensitivity' plays a central role. Community sensitivity reflects people's collective concern about their livelihood versus environmental health. When it is low, people give more weight to economic considerations and prefer to develop or control the basin, and when it is high, they give more weight to environmental considerations and prefer to protect or restore the basin. Community sensitivity depends negatively on the memory of flooding – the larger and more recent the flood, the lower community sensitivity – and positively on the degree of environmental degradation – the more degradation, the higher community sensitivity. Flooding and environmental degradation in turn depend on the regulation and restoration works that have already been executed and on external drivers, such as precipitation.

In the Kissimmee model community sensitivity differs between the urbanised upstream basin and the rural downstream basin, and overall community sensitivity is calculated as the weighted average of the two. In the Dommel basin one could distinguish between the flood-prone areas and the higher grounds. Using a limited number of equations, a coupled human–water model could be constructed that explains the decision to either regulate or restore. Given the number of parameters – six in the Kissimmee model – and the limited availability of societal data, it should be possible to achieve a reasonable fit.

Community sensitivity can be measured indirectly using newspapers articles. This involves 1) selecting one or more relevant newspapers; 2) sampling articles from these newspapers; 3) analysing quantitatively the coverage of different water–related themes; and 4) assessing qualitatively the economic or environmental 'tone' of the newspaper articles (Wei et al., 2017). Another source that could be used are children's books. A few years ago I analysed 89 children's books on flooding published in the Netherlands in the 20th Century (Mostert, 2015). The analysis showed that many books published after 1970 are critical about technology and approach nature not only as something that should be controlled, but also as something that should be protected. This indicates an increase in community sensitivity in the Netherlands around 1970. But if that is correct, why were the first river restoration works in the Dommel basin executed only around 1990? Moreover, why did the

river regulation works start in 1875 and not much earlier? To answer these questions, it is necessary to have a closer look at the developments in the Dommel basin.

## 4.2 Human activities

Several human activities have had a significant impact on the hydrology of the Dommel basin. As to flooding, we can mention four. First, already in 1818 there were complaints that the water millers maintained a too high water level, thus causing flooding upstream. This remained an issue until at least the 1930s (Deckers, 1927; Roeffen, 1963).

Secondly, in the 1840s 4,000 ha of water meadows were constructed upstream in Belgium. These were flooded in winter with water that was rich in Calcium and minerals in order to improve the fertility of the soil. The water was imported from the Meuse via a newly constructed shipping canal, but in spring the surplus water was discharged onto the Dommel, thus overcharging the river.

Thirdly, in the 19th and especially the first decades of the 20th Century large tracts of heath and moorland were converted into forests and agricultural land (Crijns and Kriellaars 1987, 1992; see Fig.2). This often involved the construction of field ditches, which increased peak runoff from the land. In 1873, the percentage of 'wild lands' in the municipalities in the southern parts of the basin still ranged from 25% to 75% (Crijns and Kriellaars 1987, 344), whereas today there is little left.

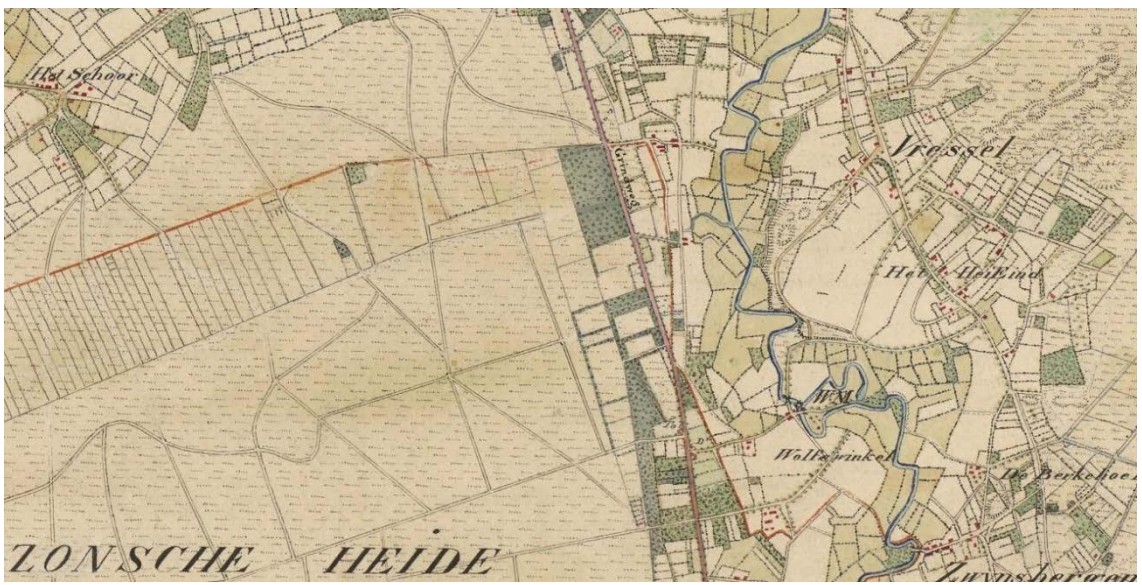

**Figure 2: Part of the Dommel basin in 1837, showing meadows along the river (light green), arable land (white) and large tracts of heath ('heide'). The Zonsche Heide is now mostly arable land, with forests and new residential areas in the south. 'WM' indicates a water mill. (National Archives, topographical map of the area around Boxtel and Sint Oedenrode (detail), toegang 4.TOPO, inv. nr 9.169)**

Fourthly, the basin became more urbanised. The population of Eindhoven, which in 1842 was only a small market town with 3,000 inhabitants (Van der Aa, 1842), grew to 46,000 inhabitants in 1920, 113,000 in 1940 and 227,000 presently. The

population of Tilburg grew in the same period from 14,000 to 194,000 inhabitants. The result was an increase in hard surfaces and in peak runoff.

To cope with the increasing flooding problems, several river regulation works were executed. Between 1875 and 1893, several small river bends were cut off, the sluices of several water mills were enlarged, and other obstacles were removed

(Bongaerts, 1919, 124–128). Between 1907 and 1911, the Drongelens Canal from Den Bosch to the Meuse was dug, which gave the Dommel an extra outlet. Following major flooding in 1917, plans were developed to divert water out of the basin (Bongaerts, 1919; Roeffen, 1963, 95–97), and between 1931 and 1941 the Dommel near Eindhoven was connected to the Wilhelmina Canal (completed in 1923). Moreover, between 1933 and 1936 a large river bend going through Boxtel was cut off.

The last wave of river regulation works started already before the Second World War but peaked in the 1960s and early 1970s (e.g. Lohman, 1963). In this period many land re-allotment projects were undertaken to reduce the number of land plots and increase efficiency in agriculture. As part of these projects, many new drainage ditches were dug and the receiving tributaries of the Dommel were straightened. Together with the ongoing urbanisation, these works further increased peak discharges onto the Dommel, and consequently plans were developed to channelize the Dommel itself.

The change from regulation to restoration was not an abrupt one. Increasingly, landscape and nature aspects were considered in land re-allotment projects (Crijns, 1998). Moreover, the plans to channelize the Dommel were only partially implemented. For the stretch between Eindhoven and Boxtel an alternative was developed involving less regulation. This plan was accepted by the water board, on the condition that the 500 to 600 ha that would continue to experience regular flooding would be bought from the farmers by nature organisations. The stretch downstream of Boxtel, however, was channelized in

1971–1972. This even led to questions in Parliament (Proceeding of the Second Chamber of Parliament 1970–1971, 2738–2748).

Actual river restoration started around 1990 (Didderen et al., 2009). Nowadays, river restoration works are undertaken as part of the implementation of the European Water Framework Directive (Mostert, 2003; Junier, 2017). Examples include re–meandering, the reconstruction of banks, and the removal of obstacles for fish migration. For the period 2016–2021 Water

Board Dommel plans to improve 93 km of watercourses (Waterschap De Dommel, 2015).

## 4.3 Actors

To understand all these activities, we need to identify the major actors and the major factors influencing them. Probably the most important group of actors were the farmers and the owners of the agricultural land. They contributed to the flooding problems by converting 'wild lands' into agricultural land, and many of the river regulation works were undertaken primarily

in their interest. Moreover, they had close connections with regional politics and with the water board.

The history of agriculture in the area has been described in great detail (Crijns and Kriellaars 1987, 1992; Crijns 1998). Around 1800 most farms were small and the farmers were poor and un-educated. Although agricultural development got

much attention, few wild lands were actually converted because of a lack of manure and other fertilizing substances. After 1840 there was a small increase in land conversion as a result of a legal change that decreased the tax burden on new agricultural lands, but from 1885 to about 1900 land conversion was practically zero as a result of a global crisis in agriculture. After 1900 it peaked. Not only had prices for agricultural produce improved, artificial fertilizers had also been introduced, which meant that nutrients were no longer a limiting factor. In addition, improvements in transportation provided better access to markets; agricultural education had been introduced; and farmers had started to organise themselves, locally in cooperatives and regionally in the North Brabant Christian Farmers Association (NBC). The NBC was closely associated with the Roman Catholic Church and with politics. Nowadays, the successor of the NBC, LTO, is no longer associated with the Church and its influence on politics has decreased, but the agricultural sector is still very well organised. On the Water Board Dommel three of the thirty seats are reserved for agriculture, two other board members are farmer as well, and one member has worked for an agricultural organisation.

The second group of actors is the non-agrarian population of the basin. As discussed in the previous section, their number increased drastically. On top of that, after the First and especially the Second World War, working hours decreased and paid holidays were introduced. The population of the ever larger cities got time to actually visit the basin and enjoy the nature and landscape (cf. Noordbrabantsch dagblad 13 October 1941; Maas, 1963, Thijsen, 1963). Already before the Second World War there were protests against the river regulation works, e.g. by the Dutch Association for the Protection of Natural Monuments, established in 1905, and sometimes plans for re-allotment were modified to accommodate the interests of nature and landscape. Yet, during the economic crisis of the 1930s and the Second World War and in the first post-war years, agricultural interests prevailed. As one journalist put it in 1941, 'what weighs most should be given most weight' (Noordbrabantsch dagblad, 11 August 1941). In the 1960s, this was still the policy of the water board (e.g. De Tijd De Maasbode, 30 June 1961; Lohman, 1963).

The third group of actors were the water millers. With the introduction of steam power, the economic value of the water mills decreased drastically. The province of North–Brabant and since 1907 the water board could regulate water levels (Deckers, 1927, 205–210), but they were not allowed to make the enjoyment of private mill rights impossible (Heemskerk, 1992). Consequently, the water board and some municipalities started to buy water mills or the mill rights only. Nowadays, nine water mills remain in the Dutch part of the basin, seven privately owned and two owned by the municipality of Eindhoven.

The fourth key actor is Water Board Dommel. Water boards have a long history in the Netherlands, but mostly in the polder areas (Van de Ven, 2004; Mostert, 2012, 2017b). In the Dommel basin there were only a few small water boards in the downstream part of the basin, which maintained and financed local dykes. In 1856, however, the Province of North–Brabant took the initiative to set up a regional water board that could execute river regulation works and tax the beneficiaries. This took seven years (Roeffen, 1963). Until 1921 the water board included only the flood-prone land in the basin, some 4,300 ha. in total (Bongaerts, 1919, 129). Only the owners of this land were charged for the expenses of the water board and could elect the board members. In 1921, the board was extended to cover the whole basin, but the higher grounds in the basin had

to pay a reduced rate. Moreover, houses and other buildings were rated for the first time, through precepts on the municipalities. The municipalities also got representatives on the water board.

The second major change was in 1950, when the water board got the task to treat wastewater. Representation of the municipalities on the board was increased to one-fifth of the seats and industry also got one-fifth of the seats. As discussed, this did not immediately lead to more attention for the 'urban interests' nature, landscape and recreation. Things really started to change after 1970. Following a controversy over the closure of the Eastern Scheldt estuary in the 1970s, a new national policy was introduced in the 1980s, called 'integrated water management'. This policy emphasised the ecological aspects of water systems (Ministerie van Verkeer en Waterstaat, 1985, 1989; Disco, 2002; Mostert, 2006). It became leading for the Dutch water boards.

The fifth group of actors are various other government bodies, such as the Province of North–Brabant, the municipalities in the basin, national government, and the European Union. The provinces can establish, regulate and change water boards, subject to approval by national government. Moreover, they can give subsidies.

The Province of North–Brabant often consulted the municipalities in the basin on water management issues. Around 1800, the municipalities had become the owner of most of the wild lands in the basin (Leenders, 1987). In later years they became responsible, among others, for the construction and maintenance of sewers and for spatial planning.

National government can adopt general legislation on for instance the water boards and agriculture. Moreover, they are responsible for works on the main rivers, such as the Meuse, and can offer subsidies for other water management works. In addition, they are responsible for international relations. In 1863, they concluded a treaty with Belgium on water allocation of the Meuse River. This treaty allowed Belgium to continue discharging the surplus water of its water meadows onto the Dommel against payment of 250,000 Belgian Francs (118,000 guilders) as damage compensation.

The European Union became very important for water management after 1970. The European Union adopted several water directives, such as the Water Framework Directive, which are binding upon the EU Member States. Moreover, the Common Agricultural Policy of the EU indirectly influences water use and quality.

Like the other actors, the different governmental bodies have changed a lot since 1800. Initially, they were like committees from the ruling class, but gradually elections became more important and in 1917 universal suffrage was introduced for men and in 1919 for women. Moreover, their staff has increased drastically. According to some, they have become the 'fourth power', in addition to the legislative, executive and judicial branch of government (Crince le Roy, 1969).

## 4.4 Factors

An important factor explaining the river regulation works is financing. The different river regulation works were only partially paid by those benefitting directly. The total costs for the first regulation works (1875–1893) were 732,312 guilders, to which Belgium contributed 118,000 guilders and the province and the State each 145,000 guilders. This left 324,312 guilders for the water board to cover, which proved a very heavy burden for the small water board. The Drongelens Canal

(1907–1911) was financed completely by the State. The costs of the works near Eindhoven (1931–1941), including some improvements for shipping, were 940,000 guilders, of which the now bigger water board covered 280,000 guilders only. Subsidies for the works in the post-war period differed, but they were typically in the order of 75%.

Many regulation works could not have been executed without subsidies, and one can doubt whether they were all
economically justified. This is hard to tell as there were also indirect benefits for the broader community, such as stimulation of the regional economy and employment, which are hard to quantify. Most likely, non-economic values played a role as well. These include the 'causation principle', according to which those contributing to a problem should contribute to its solution. This principle is embedded in the Dutch civil code, which states that owners of lower grounds have to accept the water coming from the higher grounds, but the owners of higher grounds are not allowed to make matters worse for the
lower grounds. Arguably, land conversion did make matters worse for the lower grounds, and hence all lands in the basin had to contribute to the costs the water board made (Mansholt, 1941; Schilthuis, 1960).

The second non–economic value is community and solidarity. The whole basin, the Province of North–Brabant and even the whole of the Netherlands could be seen as a community, and in a community members support each other according to their possibilities (cf. Mostert, 2017a). It is hard to assess how important exactly community and solidarity were. Traditionally,
water management in the Netherlands is based on the benefit principle and to some extent the causation principle (Schilthuis, 1960; Mostert, 2017b). Yet, mutual support, especially during and after flood disasters, is also a cultural value, as witnessed by the many children's books dealing with flooding (Mostert, 2015).

If we turn our attention to the activities that increased flooding, the picture is mixed. Economic factors were very important drivers for land conversion and urbanisation, but political and institutional factors played a role as well. Examples of the
latter include the international relations between Belgium and the Netherlands and the legal change in 1840 that decreased the tax burden on new agricultural land. Technological change was important too, most notably the introduction of artificial fertilisers.

The restoration works and the preceding protection efforts can be explained very well in terms environmental values, but these values did not depend on the state of the environment in the basin only. In addition, the changing composition of the
basin's population and their changing interests played a very important role. Moreover, how well the different interests were represented depended on institutional factors, such as the rules concerning representation on the water board. Additional factors may have been the role of individuals and their networks and the strategies they employed. The research so far has only resulted in some hints in this direction concerning for instance the role of Mr. Vosters, chairman of the water board from 1952 to 1983 and of the Dutch Association of Water Boards from 1972 to 1981 (Proceeding of the Second Chamber of
Parliament 1970–1971, p. 2742).

## 4.5 Case study conclusion

The Dommel case study has shown that the pendulum swing from water resources development and control to protection and restoration may not consist of two phases, but of three: 1) development and control (in the Dommel basin until 1970); 2) protection (between 1970 and 1990); and 3) restoration (after 1990). Moreover, the case study has shown that changes in 'community sensitivity' do not always offer a full explanation of the pendulum swing. As has also been discussed by Roobavannan et al. (2017), a change in the composition of the community may be more influential than the state of the river basin. On top of this, institutional factors can play a crucial role. Institutions such as rules on the composition of water boards influence how well the different segments of the community and their interests are represented.

Community sensitivity can be defined either as environmental concern (cf. Elshafei et al, 2014, 2144–2145) or as the balance of environmental concerns and economic concerns (Chen et al., 2016, 1229). A first point to note is that 'environmental concern' may mean different things. It may mean either concern for the economic damage that environmental change may bring (e.g. the economic damage of flooding caused by land conversion), which in the end is an economic concern, or concern for the environment itself (loss of biodiversity, landscape values). Both concerns were present in the Dommel basin. Secondly, what is missing from the concept are community values. As discussed, many of the river regulation works in the Dommel basin would not have been possible without financial contributions from the province, the State, and inhabitants of the basin that did not benefit directly. The simplest explanation for these contributions is a sense of community at different levels. How strong this sense of community was, whether it was based on a shared identity and emotional attachment or on reciprocity (a tacit understanding that any support will be reciprocated), and what role lobbying activities played, are matters that still await further study. Meanwhile, it is safe to conclude that financial arrangements are very important for understanding how water management works.

The Dommel case study clearly shows that the co-evolution of water and humans cannot be understood completely within the confines of individual river basins. This is not a new insight, yet it is not incorporated in the Kissimmee model.

If we compare the conclusions of the Dommel case study with previous socio–hydrological case studies (Wescoat, 2013; Gober and Wheater, 2014; Kandasamy at al., 2014; Liu et al., 2014), we can note that some are largely new, especially the ones on financial arrangement and community, while for other conclusions the Dommel case study offers additional support and more or different detail. Outside of socio–hydrology none of the conclusions are completely new, yet also in this case the Dommel case study offers additional support and more or different detail. Many of the conclusions could be drawn only because older literature on the Dommel or social science literature sensitised the author to the issue (e.g. financial arrangements, mentioned by Decker, 1927, and community, mentioned by Putnam, 2000, and Pretty and Ward, 2001). This points to the importance for socio–hydrology of reading widely.

Future research on the Dommel basin could model the impact of land conversion, urbanisation and river regulation on the flooding. Moreover, there is a need for more detailed research on one or two specific activities, such as the channelization of the Dommel in 1971–1972 or the first river restoration projects. This could help to find out more about the role of

individuals, social networks and political strategies. In addition, the importance of community values can be studied more. One indicator of this is memberships in social organisations (Elshafei et al., 2014; cf. Putnam, 2000). Alternatively, one could analyse the arguments used in discussions on water management (Mostert, 2012) or conduct a content analysis of newspaper articles or even children's books (see sect. 4.1). And last–but–not–least, it would be interesting to compare the Dommel basin with other basins, focusing on institutional and financial arrangements, the role of community, and their interrelation with changes in the hydrology. This calls for new case study research, but in addition the existing literature could be mined much more.

## 5. Discussion

Considering the Dommel case study and the Kissimmee model, what can we say about the best approach for socio–hydrology: case study research or coupled modelling? What are their relative advantages and disadvantages, can they be combined, and do they really differ?

To start with the last question: they do. The difference is the difference between qualitative and quantitative approaches. It is not simply a matter of words (or 'narrative') versus numbers (Creswell, 2014, 4). Rather, the aims and strategy differ. Qualitative research aims to understand the unique characteristics of individual cases. It has an inductive character, starting with observations and using theory to make sense of these observations. Quantitative research, on the other hand, aims to identify general relations, even when first individual cases are studied (or modelled). It has a deductive character, starting and ending with theory. The difference between the two, however, is not absolute, and much research falls somewhere between these two extremes.

Case studies have many strong points when it comes to improving understanding of long–term developments. They allow for the use of all types of data, including qualitative data. They are flexible, allowing the researcher to move from hydrological issues to other issues and from the basin level to higher levels without having to develop a global model of everything. Moreover, while models require that variables and many relations are specified in advance (Mount et al., 2016), this is not so for case studies. Consequently, case studies are more likely to reveal completely new variables and relations one did not think of before. Case studies also do not require the assumption of stationarity and are well suited for studying how systems and system behaviour change. Furthermore, they can go into much more detail than models. And last but not least, case studies offer good possibilities for cooperation with disciplines that are not used to (numerical) modelling.

Since qualitative case studies can be much more detailed than coupled models, they can offer far more levers for policy; obviously, models that do not include management structures and processes cannot offer any guidance on these issues. A potential weakness of case studies is generalisation, but this weakness can be addressed by selecting cases carefully and comparing between case studies, (sect. 3). A more intractable weakness of qualitative case studies is that they cannot be used for making quantitative predictions. It has to be acknowledged, though, that coupled models do not fare much better in this

respect, at least not for the long term and if by 'prediction' we imply a minimum level of certainty (cf. Srinivasan et al., 2017). For example, which coupled model could have predicted the introduction of artificial fertilisers in the Dommel basin? Coupled human–water systems are complex and to cope with this complexity compromises are inevitable (Levins, 1966; Troy et al., 2015). One option is to sacrifice generality to realism and precision, and for instance limit the geographical,

temporal and thematic scope of the model. This is effectively the strategy of demand–driven modelling (Garcia et al., 2016). Another option is to sacrifice realism to generality and precision. This is what may happen if one makes models too complex to be calibrated properly or if one leaves out crucial processes in order to limit complexity. A third option is to sacrifice precision to realism and generality. Uncalibrated toy models fit in this strategy, but if one is willing to sacrifice precision, qualitative case study research is a good option too.

A strong point of coupled models is that, even though they cannot predict the (far) future very well, they can be used for generating quantitative scenarios or *possible* futures (plural), using different assumptions concerning for instance climate change and economic growth. These scenarios can then be used for developing robust and flexible management strategies (e.g. Pahl–Wostl et al., 2008; Haasnoot et al., 2013). Moreover, coupled models can be used to explore the *possible* effects of policy measures (such as an insurance system for flood damage: Grelot and Barreteau, 2012). And last–but–not–least,

coupled toy models can be used to generate *possible* explanations of observed phenomena, provided the model is actually played with, e.g. different parameter values are tried out. A good example is Yu et al. (2017).

Qualitative case studies and modelling can be combined in different ways. One could start with a case study and use the results for developing a realistic coupled model, as was done for instance by Kandasamy et al. (2014) and Van Emmerik et al. (2014). This is a sound approach, but it will generally not be possible to keep all the richness of the case study without

making the model too complex. Secondly, case study research could follow modelling and could then be used to help interpret the results, validate the model, or test specific hypotheses. And thirdly, modelling can be part of a larger qualitative case study and can then be used to determine the impact of human activities (non-coupled models) or to generate possible explanations (coupled toy models).

The research approaches and methods that can be used in socio–hydrology are not limited to qualitative case study research

and modelling. Other options include large–N statistical studies (e.g. Hornberger et al., 2015), which seem very promising for comparative socio–hydrology, and survey research, for instance on how farmers actually react to hydrological change (Sanderson and Curtis, 2016) or on what actually determines environmental concern: personal characteristics or local environmental problems (Hannibal et al., 2016). The reason why this article focused on qualitative case study research and coupled modelling and is that coupled modelling is the most popular approach in socio–hydrology and qualitative case study

research is a contrasting approach with many potential benefits.

It would be possible to leave the discussion here and conclude that 1) qualitative case study research and coupled modelling both have strong and weak points, 2) they can be combined to some extent, and 3) there are more approaches that can be used in socio–hydrology, but these are not the topic of this article. Yet, I would like add one more conclusion: 4) case study research deserves more attention in socio–hydrology. There are few socio–hydrological case studies, but they can help to

make sense of the real–life complexities of humans and water and it is possible to generalise on the basis of them. Moreover, they are a good vehicle for bringing together literature, experts and insights from many different disciplines. Hence, they should receive more attention.

*Competing interests*. The author declares that he has no conflict of interests.

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
