# Peer review of "An alternative approach for socio-hydrology: case study research"

_Hydrology and Earth System Sciences, 2017_

## Referee Comment (RC1) · Anonymous Referee #1 · 23 Jun 2017

I welcome this paper, it attracts attention to an area that has not been well covered in the literature.

But I believe the paper is based on a false or ill-informed premise. By calling it "an alternative approach", the author claims he is proposing something that was not previously known to people working in socio-hydrology. This is false - he might have got this impression from selective reading of the literature. I see that much of the literature he cites is work on the "levee effect". It is true that this research started from conceptual modeling, which were not necessarily inspired from actual case studies or data arising from them.

In reality, there is a branch of socio-hydrology, that on human-environment competition for water in the context of irrigated agriculture. Most/all of these based on real case

studies. These include the work of Kandasamy et al. (HESS) and Liu et al. (HESS, China). Early papers were data-based historical narratives, which generated ideas and hypotheses, which were followed by modeling studies (van Emmerik et al, HESS, Liu et al., HESS). There were other studies like Elshafei et al (HESS, WRR) which were inspired by previous historical studies/narratives. Chen et al. (WRR, Florida) is another work which combined data analysis and modeling in a real case study. The work of Srinivasan (HESS) was a study that indeed a case study focused on a city, combined with modeling. It also included multiple social actors.

These studies were followed by the review paper of Sivapalan and Bloeschl (2015) who provided guidance to socio-hydrology studies, one element of which was indeed the generation of narratives based on real case studies and expressing these in terms of unexplained phenomena (either local or universal) which will then generate the hypotheses to be explored through modeling studies.

So this is why I said this manuscript is based on a false or ill-informed premise.

Having said that, I will welcome it if the author takes the idea of case studies to expand into territory not well covered in previous studies, including modeling studies. I do completely accept the point that in previous studies the social aspect may have been lumped into one abstract concept of a social variable (flood memory in levee effect papers, community sensitivity). It can be disaggregated into different parts of the social system, which might also include governance systems.

If this is what he wants to do, then by all means illustrate this through an example case study, and demonstrate either conceptually, through modeling or through data analysis, why any conclusions one makes can be seriously impacted by the lumping.

Unfortunately, even when the author presents a case study, the paper goes back to theoretical and philosophical issues and does not deliver anything new that I did not know already. I do not like to more criticism, discussion and philosophy - I want real case studies from which I can learn something that I do not know already. I guess that

might mean a totally different approach to presenting this paper, more data analyses or building of conceptual models inspired by a real place/case study - this calls for major revision

---

## Referee Comment (RC2) · Anonymous Referee #2 · 27 Jun 2017

Mostert argues for an increased focus on detailed case study research in the field of socio-hydrology. He reviews the socio-hydrology literature noting the modeling focus. He then presents case study research as an alternative approach and compares the advantages and disadvantages of the two approaches. As an example of case study research in socio-hydrology, Mostert describes the case of Dommel Basin, located in Belgium and the Netherlands. Mostert details the limitations of modeling and demonstrates that a diversity of approaches is needed to understand complex socio-hydrological systems. The manuscript makes an important point and the topic is of interest to Hydrology and Earth Systems Science readers but discussion of existing socio-hydrology case study research and more support of the methods, advantages, and disadvantages of the case study approach is needed. I have a series of specific

comments that, if addressed, would strengthen the paper:

1. In the discussion socio-hydrological research approaches, the author focuses on modeling which he characterizes as the dominant approach. However, there are examples of case study research in socio-hydrology and a more nuanced discussion that discussion these examples and their strengths and weaknesses is needed (Gober and Wheater 2014; Kandasamy et al., 2014; Liu et al. 2014; Treuer et al., 2017). Additionally, there are numerous examples of case study work aiming to address questions relevant to socio-hydrology that are not explicitly categorized as socio-hydrological research. Acknowledging these efforts would further illustrate the potential of socio-hydrological case study research and point researchers to related work not yet well integrated in to socio-hydrology (for a few examples see: http://sfwsc.fiu.edu/Research_Questions.html, https://www.nsf.gov/awardsearch/showAward?AWD_ID=1204685, https://www.nsf.gov/awardsearch/showAward?AWD_ID=0948914 (missing from the last list: Mini et al. 2014)).

2. On page 3, the author notes that moving beyond the scale of the river basin to incorporate factors such as trade would necessarily result in more complex models. Model complexity should correspond with the model aims not model scale, a point which both references cited make (Pande and Sivapalan 2016, p13; Srinivasan et al. 2017, p5). Please revisit this point.

3. In section 3, the author describes the case study approach. This section is central to the author's message but there are notably few references here. Further examples of case study research that demonstrates the use of alternate data types, the range of questions addressed, theory guided analysis, and case selection. In particular, the discussion of case study selection criteria is an important one that is need of expansion. While the two methodological references provided are useful, examples of each of the case study selection strategies are needed. These examples need not be from water resources if more appropriate examples are found in other fields (i.e. land use change,

energy).

4. In section 4, the author presents the case of the Dommel Basin. The addition of a case study example is welcome here but the case as currently presented is weak. A stronger example would demonstrate the general points made in section 3 (i.e. range of research questions that case studies can address, integrating different data types, selecting case(s), etc.) rather than again stating these points. Revising this case to illustrate how the case study approach enables a more nuanced understanding of the shift from development to restoration, how different data types can be combined and why this case was selected would strengthen this example. Further, continuing the comparison between the Kissemee model and the Dommel Basin case to the discussion of findings would improve this discussion.

5. The author makes the important point that many socio-hydrology articles use modeling for inference but that a diversity of approaches is beneficial. However, the author presents modeling and case studies as the only two options for socio-hydrological research. Other approaches, such as large-N statistical studies, are also not widely used in socio-hydrology and have certain advantages and disadvantages (i.e. Hornberger et al. 2015). Other approaches beyond case studies and modeling should be acknowledged and the reason for the focus on case studies clarified.

6. The author rightly notes limited generalization as one drawback of the case study approach but it is worth mentioning here that there are efforts to address this challenge in socio-hydrology through meta-analysis of case studies and by synthesizing quantitative and qualitative data from case studies (Srinivasan et al. 2012; Treuer et al., 2017).

7. On page 2 line 8, and again on page 3 line 1, the term "socio-ecological" should read "socio-hydrological."

References

Gober, P., and Wheater, H. S. (2014). "Socio-hydrology and the science–policy interface: a case study of the Saskatchewan River basin." Hydrology and Earth System Sciences, 18(4), 1413–1422.

Hornberger, G. M., Hess, D. J., and Gilligan, J. (2015). "Water conservation and hydrological transitions in cities in the United States." Water Resources Research, 51(6), 4635–4649.

Kandasamy, J., Sountharajah, D., Sivabalan, P., Chanan, a., Vigneswaran, S., and Sivapalan, M. (2014). "Socio-hydrologic drivers of the pendulum swing between agricultural development and environmental health: A case study from Murrumbidgee River basin, Australia." Hydrology and Earth System Sciences, 18(3), 1027–1041.

Liu, Y., Tian, F., Hu, H., and Sivapalan, M. (2014). "Socio-hydrologic perspectives of the co-evolution of humans and water in the Tarim River basin, Western China: the Taiji–Tire model." Hydrology and Earth System Sciences, 18(4), 1289–1303.

Mini, C., Hogue, T. S., and Pincetl, S. (2014). "Patterns and controlling factors of residential water use in Los Angeles, California." Water Policy, 1–16.

Pande, S., and Sivapalan, M. (2016). "Progress in socio-hydrology: a meta-analysis of challenges and opportunities." WIREs Water.

Treuer, G., Koebele, E., Deslatte, A., Ernst, K., Garcia, M., and Manago, K. (2017). "A narrative method for analyzing transitions in urban water management: The case of the Miami-Dade Water and Sewer Department." Water Resources Research.

Srinivasan, V., Lambin, E. F., Gorelick, S. M., Thompson, B. H., and Rozelle, S. (2012). "The nature and causes of the global water crisis: Syndromes from a meta-analysis of coupled human-water studies." Water Resources Research, 48(10), n/a-n/a.

Srinivasan, V., Sanderson, M., Garcia, M., Konar, M., Blöschl, G., Sivapalan, M., Sanderson, M., Garcia, M., Konar, M., Blöschl, G., and Sivapalan, M. (2016). "Prediction in a socio-hydrological world." Hydrological Sciences Journal – Journal des

[Figure]

Sciences Hydrologiques, Taylor & Francis, 0(0), 1–8.

---

## Author Comment (AC1) · 28 Aug 2017

1. First of all, I would like to thank the two anonymous reviewers for their reviews. They have pointed to areas where the paper should be improved or clarified.

2. In short, my paper reviews the dominant approach in socio-hydrology, coupled modelling, and proposes an alternative, qualitative case studies. Moreover, it presents a short case study to show the potential benefits of this alternative, as well as being valuable on its own. I will state this more clearly in the introduction.

3. I am aware that there are more alternatives to modelling than qualitative case studies, such as large-N statistical studies. I chose case studies as alternative to discuss, first of all, because I think it will result in a more nuanced understanding of the issues

socio-hydrology is interested in. The improved Dommel case study should show this (see point 6 below). Secondly, modelling and large-N statistical studies are both examples of quantitative approaches, whereas case studies are the quintessential qualitative approach. In the discussion section I will discuss the difference (see point 5 and 12).

4. I am also aware that there are some qualitative socio-hydrological studies. I will mention the existing studies and where appropriate use them to illustrate the points I make. For instance, I will refer to Gober and Wheater (2014) and Kandasamy et al. (2014) when I stress the importance of institutions and government policies in the Dommel case, and also identify what the Dommel case adds compared to these references.

5. The main comment of reviewer 1 is whether case studies really are an alternative to modelling. As I will mention in the discussion section, it is true that many socio-hydrological models are informed by previous qualitative case studies, yet there are some fundamental differences. Quantitative research such as modelling aims at generalisation, while qualitative research such as case study research aims at a detailed understanding of specific cases. When qualitative narratives are translated into formal models, they become more broadly applicable, but inevitably a lot of contextual information is lost. Compare for instance the qualitative description of developments in the Murrumbidgee river basin in Kandasamy et al. 2014, in HESS, with the coupled model of the same basin presented in Van Emmerik et al. 2014, also in HESS. Or compare the case description in Pande et al. 2016 in Wires Water with the model presented in that paper.

6. The review of socio-hydrology in section 2 is focused on the dominant approach in socio-hydrology, coupled modelling. I will explain that I have collected all published socio-hydrological models and analysed how society is included in these. I used other socio-hydrological literature only if it discusses modelling issues. Reviewer 2 mentions that there is a lot of literature addressing questions relevant to socio-hydrology that is not explicitly categorized as socio-hydrological. I agree, and this is an important point

to emphasise. I cannot review all this literature in one paper, but I have used a lot of this type of literature for the Dommel case study and I also use it in the introduction.

7. The biggest and most important change I plan to make is to strengthen the Dommel case study. Both reviewers pointed to the need for this. I will link the case study better to the preceding section and explain why I chose the Dommel: it is a "typical example" of a more general phenomenon, the pendulum shift from regulation and control to protection and restoration. Next, I will systematically discuss 1) the human activities that had a significant impact on the basin, and 2) the factors that can explain these activities. I will specify these factors and formulate them in general terms, so that they can inform subsequent case studies and modelling efforts. I will give more details and cite more sources.

8. I hope this addresses all comments made by reviewer 1: the premise that case study research is an alternative approach (point 5), the selective reading of the literature (point 6), and the need to strengthen the case study (point 7). The proposed changes will not make the paper less critical, but they intend to make the paper less philosophical and more methodological and empirical.

9. Reviewer 2 has some additional comments. In section 3 on case study research, reviewer 2 would like to see more references to different types of case studies and case study selection strategies. What I intend to so is to distinguish more clearly between 1) case study research in generally, and 2) the specific type of case study to study socio-hydrological questions. The main example of the latter is the Dommel case. I will look for good examples of other type of case studies as well, but it will not be possible to present them in much detail.

10. On page 3 I note that moving beyond the river basin scale and including more variables and processes will result in more complex models, "all else being equal." This is an essential proviso. As reviewer 2 notes, both references I cite mention that model complexity should correspond with the model aims, not model scale. I agree.

Still, complexity is an issue. If variables are added for which no data are available, the degrees of freedom of the model will increase. This may result in a better fit, but not necessarily for the right reasons.

11. Reviewer 2 points to efforts to generalise on the basis of case studies. I will discuss these in section 3 on case studies and come back to it in the discussion section.

12. I plan to combine sections 5 and 6, comparison and discussion, into one section called discussion. This section will discuss 1) the comparative advantages of qualitative case study research and modelling, using the Dommel case study and the Kissemee model as examples; 2) the fundamental difference between quantitative approaches such as modelling and qualitative approaches such as case study research (briefly!); and 3) ways to combine quantitative and qualitative research, including the use of case study research to inform modelling, the use of modelling to inform case studies, and means to generalise on the basis of case study research and hybrid approaches.

---

## Author Response (AR1)

**Author's point-by-point reply to reviews of: An alternative approach for socio–hydrology: case study research, Erik Mostert**

First of all, I would like to thank the two reviewers for the reviews they provided. These have helped me to improve the paper. Before going into detail, it may be useful to outline the main changes:

a. The Dommel case study has been extended and restructured. It now goes into much more detail, uses more sources, and follows the structure proposed in the previous section. Moreover, a case study conclusion has been added and the case is compared more systematically with the Kissimmee model.

b. The discussion and conclusion sections have been combined and now discuss the general issue of coupled models versus (or: and) case studies.

c. The abstract has been rewritten to better reflect the new version of the paper.

The structure of the first three sections has not changed, but some important improvements have been made. These are outlined below. Because of all the changes, including many moved text and minor editorial improvements, I do not recommend to read the marked-up manuscript version, attached to this reply..

Anonymous Referee #1

1. I welcome this paper, it attracts attention to an area that has not been well covered in the literature.

Reaction: Thanks for the appreciation.

2. But I believe the paper is based on a false or ill-informed premise. By calling it "an alternative approach", the author claims he is proposing something that was not previously known to people working in socio-hydrology. This is false (...).

Reaction: In the revised version I have tried to prevent any suggestion that case study research is completely new to socio–hydrology. My main conclusion is that case studies deserve "more" attention, which I also mention in the abstract and the introduction. Moreover, in section 2 I have added references to four previous socio-hydrological case studies, and in section 4.5 I discuss what new insights the Dommel case study brings.

3. (H)e might have got this impression from selective reading of the literature. I see that much of the literature he cites is work on the "levee effect". It is true that this research started from conceptual modeling, which were not necessarily inspired from actual case studies or data arising from them. In reality, there is a branch of socio-hydrology, that on human-environment competition for water in the context of irrigated agriculture. Most/all of these based on real case studies. These include the work of Kandasamy et al. (HESS) and Liu et al. (HESS, China). Early papers were data-based historical narratives, which generated ideas and hypotheses, which were followed by modeling studies (van Emmerik et al, HESS, Liu et al., HESS). There were other studies like Elshafei et al (HESS, WRR) which

were inspired by previous historical studies/narratives. Chen et al. (WRR, Florida) is another work which combined data analysis and modeling in a real case study. The work of Srinivasan (HESS) was a study that indeed a case study focused on a city, combined with modeling. It also included multiple social actors. These studies were followed by the review paper of Sivapalan and Bloeschl (2015) who provided guidance to socio-hydrology studies, one element of which was indeed the generation of narratives based on real case studies and expressing these in terms of unexplained phenomena (either local or universal) which will then generate the hypotheses to be explored through modeling studies.

Reaction: I have tried to make clear that I do not focus specifically on the levee effect. In the new introduction I make clear that my review of the socio-hydrological literature in section 2 focuses on the coupled models that have been made and I have also added "modelling" in the heading of section 2. These models include not only models of the levee effect but also models of human-environment competition for water and the pendulum swing from development and control to protection and restoration. Section 4.1 discusses one model of the pendulum swing in detail (Chen et al.). Moreover, in section 5 I have added a discussion of how coupled models and case studies can be combined, mentioning Kandasamy and Van Emmerik et al. as an example.

4. Having said that, I will welcome it if the author takes the idea of case studies to expand into territory not well covered in previous studies, including modeling studies. I do completely accept the point that in previous studies the social aspect may have been lumped into one abstract concept of a social variable (flood memory in levee effect papers, community sensitivity). It can be disaggregated into different parts of the social system, which might also include governance systems. If this is what he wants to do, then by all means illustrate this through an example case study, and demonstrate either conceptually, through modeling or through data analysis, why any conclusions one makes can be seriously impacted by the lumping.

Reaction: This is indeed what I intend to do by means of the Dommel case study. I have completely rewritten the case study and expanded it a lot. Section 4.5 provides an answer to the question "why any conclusions one makes can be seriously impacted by the lumping." In one sentence: crucial variables and processes may be missed. As discussed in section 2, a good match between model results and data does not necessarily mean that all crucial variables and processes have been included. It could be that crucial variables and processes did not change in the area and period covered, it could be that the model includes many variables for which no data are available and consequently has many degrees of freedom, and it could be a combination of both.

5. Unfortunately, even when the author presents a case study, the paper goes back to theoretical and philosophical issues and does not deliver anything new that I did not know already. I do not like to more criticism, discussion and philosophy - I want real case studies from which I can learn something that I do not know already. I guess that might mean a totally different approach to presenting this paper, more data analyses or building of conceptual models inspired by a real place/case study - this calls for major revision

Reaction: I hope the extended discussion of the Dommel can count as a real case study. Moreover, I have limited criticism, discussion and philosophy to the final discussion section, which discusses the relative pros and cons of case study research and coupled modelling, their fundamental differences, and the possibilities to combine them.

Anonymous Referee #2

6. Mostert argues for an increased focus on detailed case study research in the field of socio-hydrology. He reviews the socio-hydrology literature noting the modeling focus. He then presents case study research as an alternative approach and compares the advantages and disadvantages of the two approaches. As an example of case study research in socio-hydrology, Mostert describes the case of Dommel Basin, located in Belgium and the Netherlands. Mostert details the limitations of modeling and demonstrates that a diversity of approaches is needed to understand complex sociohydrological systems. The manuscript makes an important point and the topic is of interest to Hydrology and Earth Systems Science readers but discussion of existing socio-hydrology case study research and more support of the methods, advantages, and disadvantages of the case study approach is needed. I have a series of specific comments that, if addressed, would strengthen the paper:

Reaction: I appreciate the overall assessment. I agree there were areas for improvement, and I thank the reviewer for pointing these out. Below, I indicate how I have addressed the more specific comments.

7. In the discussion socio-hydrological research approaches, the author focuses on modeling which he characterizes as the dominant approach. However, there are examples of case study research in socio-hydrology and a more nuanced discussion that discussion these examples and their strengths and weaknesses is needed (Gober and Wheater 2014; Kandasamy et al., 2014; Liu et al. 2014; Treuer et al., 2017).

Reaction: In section 2 I now mention four socio-hydrological case studies explicitly, and in section 4.5 I briefly compare the Dommel case study with these studies.

8. Additionally, there are numerous examples of case study work aiming to address questions relevant to socio-hydrology that are not explicitly categorized as socio-hydrological research. Acknowledging these efforts would further illustrate the potential of socio-hydrological case study research and point researchers to related work not yet well integrated in to sociohydrology (for a few examples see: http://sfwsc.fiu.edu/Research_Questions.html, https://www.nsf.gov/awardsearch/showAward?AWD_ID=1204685, https://www.nsf.gov/awardsearch/showAward?AWD_ID=0948914 (missing from the last list: Mini et al. 2014)).

Reaction: I have tried to acknowledge the importance of work not yet well integrated in to socio-hydrology. The new case study conclusion (section 4.5, last paragraph) emphasizes the importance of older literature on specific basins and of water management and social science more generally, and mentions a few examples. Other examples can be found throughout the paper, e.g. in the very first paragraph. In the penultimate sentence of the paper I conclude that case studies are a good vehicle for bringing together literature, experts and insights from many different disciplines.

9. On page 3, the author notes that moving beyond the scale of the river basin to incorporate factors such as trade would necessarily result in more complex models. Model complexity should correspond with the model aims not model scale, a point which both references cited make (Pande and Sivapalan 2016, p13; Srinivasan et al. 2017, p5). Please revisit this point.

Reaction: I have changed the formulation of the sentence: "Unless one simplifies in other respects, this actually results in more complex models." I agree that model complexity should correspond with model aims, but data availability is a limiting factor, and if one incorporates additional factors, other factors should be excluded if one does not want to increase complexity.

10. In section 3, the author describes the case study approach. This section is central to the author's message but there are notably few references here. Further examples of case study research that demonstrates the use of alternate data types, the range of questions addressed, theory guided analysis, and case selection. In particular, the discussion of case study selection criteria is an important one that is need of expansion. While the two methodological references provided are useful, examples of each of the case study selection strategies are needed. These examples need not be from water resources if more appropriate examples are found in other fields (energy).

Reaction: Referencing has been improved. I have added references to examples of case selection strategies, as well as some methodological references that I find particularly useful. Concerning the data sources that can be used I refer to the Dommel case study, which in effect uses many. The purpose of the section is not to discuss case study research extensively, but to introduce the approach and specify it for socio-hydrology, e.g. by proposing three central questions. The central section in the paper is the expanded section 4: the Dommel case study.

11. In section 4, the author presents the case of the Dommel Basin. The addition of a case study example is welcome here but the case as currently presented is weak. A stronger example would demonstrate the general points made in section 3 (i.e. range of research questions that case studies can address, integrating different data types, selecting case(s), etc.) rather than again stating these points. Revising this case to illustrate how the case study approach enables a more nuanced understanding of the shift from development to restoration, how different data types can be combined and why this case was selected would strengthen this example. Further, continuing the comparison between the Kissemee model and the Dommel Basin case to the discussion of findings would improve this discussion.

Reaction: The Dommel case study has been completely rewritten. It now contains references to several data sources: old maps, the archives of government bodies, old census data, topographical descriptions, newspaper reports, old consultancy reports and other studies. The structure of the Dommel case study has been brought in line with what is proposed in section 3, with separate sub-sections for each central question (4.2: main activities, 4.3: main actors, and 4.4: main factors). The Dommel basin is first introduced as a "typical example" of the pendulum swing. This is the first option for selecting cases mentioned in sect. 3. Moreover, the comparison between the Kissimmee model and the Dommel case study is continued into section 4.5 with the case study conclusion. Information that was repetitive has been removed.

12. The author makes the important point that many socio-hydrology articles use modeling for inference but that a diversity of approaches is beneficial. However, the author presents modeling and case studies as the only two options for socio-hydrological research. Other approaches, such as large-N statistical studies, are also not widely used in socio-hydrology and have certain advantages and disadvantages (i.e. Hornberger et al. 2015). Other approaches beyond case studies and modeling should be acknowledged and the reason for the focus on case studies clarified.

Reaction: A paragraph has been added in the final section that discusses other approaches and gives some examples, including Hornberger et al. (2015). Moreover, it states that "the reason why this article focused on coupled modelling and qualitative case study research is that coupled modelling is the most popular approach in socio–hydrology and qualitative case study research is a contrasting approach with many potential benefits."

13. The author rightly notes limited generalization as one drawback of the case study approach but it is worth mentioning here that there are efforts to address this challenge in socio-hydrology through meta-analysis of case studies and by synthesizing quantitative and qualitative data from case studies (Srinivasan et al. 2012; Treuer et al., 2017).

Reaction: Both references have been added in section 3: Srinivasan et al. (2012) as an example of a meta-analysis of published case studies, and Treuer et al (2017)as an example of the use of theory in case studies.

14. On page 2 line 8, and again on page 3 line 1, the term "socio-ecological" should read "socio-hydrological."

This has been corrected.

[revised manuscript text omitted]

---

## Author Response (AR2)

Reviewer number one had a few editorial questions/ suggestions. Here is how I have addressed them:

1. Is Water board Dommel the official name of the board? If so, 'board' should also be capitalized. If not, the 'Dommel water board' would be the correct syntax.

Reply: It is. Whenever 'water board' is part of a proper name, I have capitalised 'board' as well.

2. In the list of potential data source I would cut 'old' from the description of maps, surveys, laws, etc. as both old and current maps, etc. are useful data sources.

Reply: Done

3. On page 11, line 25 clarify the meaning of 'they'. It is not clear if they refers to provincial, local and / or EU level governmental bodies.

Reply: 'They' refers to the subject of the previous sentence, 'the different governmental bodies', hence to all governmental bodies existing at the time.

4. Overall, the writing is a bit informal, though it is clear. I recommend the author review the manuscript with a focus on writing style. The last paragraph jumped out to me as particularly informal.

Reply: I have tried to use plain and active language that is understandable for educated persons whatever their disciplinary background. The result may be a bit informal. However, I have reviewed the whole manuscript and replaced some colloquial words and expressions (e.g. 'a lot of' by 'much'). In the last paragraph, first sentence, I have replaced 'I could' by 'it would be possible to'. In addition, I have made a few other minor adjustments to improve flow (e.g. split a sentence in two or used a different conjunction) and corrected some typos.